# Correlation between Carbonic Anhydrase Isozymes and the Evolution of Myocardial Infarction in Diabetic Patients

**DOI:** 10.3390/biology11081189

**Published:** 2022-08-08

**Authors:** Sorina Magheru, Calin Magheru, Florin Maghiar, Liliana Sachelarie, Felicia Marc, Corina Maria Moldovan, Laura Romila, Anica Hoza, Dorina Maria Farcas, Irina Gradinaru, Loredana Liliana Hurjui

**Affiliations:** 1Department of Medical Disciplines, Faculty of Medicine and Pharmacy, University of Oradea, 410073 Oradea, Romania; 2Departament of Preclinical Disciplines, Apollonia University, 700511 Iasi, Romania; 3Department of Medical Disciplines, Faculty of Medicine and Pharmacy, “Grigore T. Popa” University of Medicine and Pharmacy, 700115 Iasi, Romania

**Keywords:** acute myocardial infarction, diabetes, lactic acid, carbonic anhydrase

## Abstract

**Simple Summary:**

Heart disease in diabetics presents distinctive characteristics both anatomically and physiopathologically compared to non-diabetics. In people with diabetes, high blood pressure has a high incidence (approximately one-third of diabetic patients have high blood pressure) and is a risk factor for diabetic macro- and microvascular complications. The correlation of these parameters could represent early markers of the prognosis and evolution of diabetic patients with acute myocardial infarction and their routine determination could be included in the biological algorithm of acute myocardial infarction, but understanding of this aspect must be deepened in the future. The results showed that diabetic patients develop acute myocardial infarction more frequently, regardless of age. The level of the enzymes of myocardial necrosis was higher in diabetics compared to non-diabetics, and acute coronary syndrome occurs mainly in diabetics with inadequate metabolic balance. Our research may provide useful information for the medical community.

**Abstract:**

(1) Background: Myocardial infarction was, until recently, recognized as a major coronary event, often fatal, with major implications for survivors. According to some authors, diabetes mellitus is an important atherogenic risk factor with cardiac determinations underlying the definition of the so-called “diabetic heart”. The present study aims to establish a correlation between the evolution of myocardial infarction in diabetic patients, by determining whether lactic acid levels, the activity of carbonic anhydrase isoenzymes, and the magnitude of ST-segment elevation are correlated with the subsequent evolution of myocardial infarction. (2) Methods: The study analyzed 2 groups of 30 patients each: group 1 consisted of diabetic patients with acute myocardial infarction, and group 2 consisted of non-diabetic patients with acute myocardial infarction. Patients were examined clinically and paraclinical, their heart markers, lactic acid, and the activity of carbonic anhydrase I and II isozymes were determined. All patients underwent electrocardiogram and echocardiography analyses. (3) Results: The results showed that diabetics develop acute myocardial infarction more frequently, regardless of how much time has passed since the diagnosis. The value of myocardial necrosis enzymes was higher in diabetics than in non-diabetics, and acute coronary syndrome occurs mainly in diabetics with poor metabolic balance. Lethality rates in non-diabetic patients with lactic acid values above normal are lower than in diabetics. (4) Conclusions: Lactic acid correlated with the activity of isozyme I of carbonic dioxide which could be early markers of the prognosis and evolution of diabetic patients with acute myocardial infarction.

## 1. Introduction

Until recently, myocardial infarction was recognized as a major coronary event, often fatal, with major implications for survivors [1,2]. This paradigm has changed as a result of improved therapeutic strategies in patients with coronary heart disease and due to the methods of detecting or excluding myocardial necrosis. The introduction of techniques for measuring cardiac troponins has allowed the detection with high sensitivity and accuracy of even small areas of myocardial necrosis, changing the criteria for defining infarction [3].

As a consequence of applying the new definition, more patients with myocardial infarction will be identified and many more episodes of reinfarction will be diagnosed in patients with progressive coronary heart disease. According to the new definition, patients requiring myocardial revascularization (interventional or surgical) are at risk of additional myocardial destruction or myocardial infarction [4,5]. The detection of microinfarcts, possibly due to the new markers produced during the revascularization maneuvers, places these patients in the high risk group [6,7]. However, the patient’s long-term prognosis can be significantly improved by interventional or surgical revascularization. For example, a patient with unstable angina and severe anterior descending artery stenosis will have a much greater benefit from coronary dilation despite a small increase in cardiac troponins. The benefit will far outweigh the negative impact of periprocedural infarction.

According to some authors, diabetes mellitus is an important atherogenic risk factor with cardiac determinations underlying the definition of the so-called “diabetic heart”. Heart disease in diabetics has distinctive features from non-diabetics, both anatomically and pathophysiologically. The concept of diabetic heart muscle disease, later called diabetic cardiomyopathy, involves diffuse myocardial damage even with coronary and microangiopathic lesions being absent. The physio-pathological processes that lead to the significant increase in cardiovascular damage in patients with diabetes are represented by three important biohumoral changes: hyperglycemia, hyperinsulinemia and the increase in circulating free fatty acids [8].

Hyperglycemia is the negative factor that influences the prognosis and clinical evolution of AMI, leading to higher mortality and more severe complications. Hyperglycemia is an important factor that determines endothelial dysfunction, leading to vascular damage and microvascular obstructions, and its effect of increasing oxidative stress, inflammation and platelet aggregation is also known. Thus, hyperglycemia causes a change in protein synthesis, with increased expression of pro-inflammatory factors (IL11, IL 6, TNF alpha, fibrinogen), oxidative stress and adhesion molecules (ICAM 1, VCAM 1), all of which result in endothelial damage and the appearance of atherosclerotic lesions [9,10].

Diabetes is accompanied by an extremely high incidence of acute cardiovascular events; thus, 15–20% of patients presenting with acute myocardial infarction also have diabetes. The subsequent evolution of these patients is affected, the risk of adverse events being twice as high as in non-diabetics. The risk of cardiovascular disease is increased at blood sugar values slightly above the allowed limit and even at values above the normal limit. Hyperglycemia is associated with a poor prognosis in patients with acute myocardial infarction, regardless of the presence of diabetes [11,12].

Coronary findings in diabetes are of two types: macroangiopathy affecting the epicardial coronary vessels and microangiopathy characterized by thickening of the basement membrane of capillaries and arterioles with a diameter below 150 µm.

The clinical expression of cardiac involvement in diabetics also includes the autonomic cardiac neuropathy that appears in diabetic neuropathy and is usually associated with other types of neuropathic complications. High blood pressure has a high incidence in people with diabetes (about a third of diabetic patients have high blood pressure) and is a risk factor for the macro- and microvascular complications of the diabetic patient [13,14].

Although diabetes mellitus is frequently associated with other cardiovascular risk factors (age, abdominal obesity, hypertension, dyslipidemia), it is an independent cardiovascular risk factor that acts even in the absence of other risk factors or after other risk factors are brought under control [15,16].

Carbonic anhydrase (CA), an enzyme discovered in 1932 by Meldrum and Roughton, catalyzes one of the simplest chemical reactions, that of carbon dioxide and water [17].

The physiological relevance of CA isozymes and their possible fundamental role in the living organism derives from the discovery of some families which genetically lack cytosolic CA II [18,19]. These patients had symptoms of osteopetrosis, renal tubular acidosis and cerebral calcifications. These findings are currently included in other investigations, aimed, both at the quantitative determination of the activity of other CA isozymes in various organs and tissues of individuals lacking CA II, and in the research and study of an animal model in which one or more isozymes are completely missing [20,21].

Transport and elimination of CO_2_, regulation of pH and osmotic pressure, fluid secretion and regulation of the ion exchange process [16] are therefore the most common physiological processes involving the catalytic activity of one or more isozymes of CA. Regarding the physiological role of CA isozymes, the research team led by Prof. Puscas showed that I isozymes are involved in modulating vascular processes in the body [22,23] in which substances with vasodilating effects inhibit CA I, and vasoconstrictors activate these isozymes, while at the same time, II isozymes are involved in the modulation of secretory processes in the body, all these results materializing into a new concept of signal transmission in the cell—a theory of pH [24,25].

## 2. Materials and Methods

### 2.1. Aim of the Study

This study proposed to establish whether there was a correlation between the evolution of myocardial infarction in diabetic patients and the activity of CA isoenzymes. The presence of lactic acid and the activity of CA I and CA II isozymes were determined and the values were correlated with the magnitude of the ST segment elevation and with the subsequent evolution of myocardial infarction.

### 2.2. Materials

We studied a total of 60 patients, who were divided into two groups. All patients were hospitalized with acute myocardial infarction at the CF Oradea Clinical Hospital, over two years.

Group 1—Included 30 diabetic patients with acute myocardial infarctionGroup 2—Included 30 non-diabetic patients with acute myocardial infarction

The measurement of lactic acid and markers of cardiac necrosis (immediately after admission) and the use of the two groups of patients arose from the need to determine biological parameters that would lead us to early diagnosis. These markers could indicate a prospective unfavorable evolution of acute myocardial infarction in diabetics compared to non-diabetics, and thus allow the use of more aggressive treatment, followed by an improvement in the evolution and the quality of life of those patients. We determined the value of lactic acid, cardiac markers and the activity of CA isozymes and looked for a correlation between their levels, the size of ST-segment elevation and the subsequent evolution of patients.

We prepared a study sheet for each patient, which included data for a single hospitalization.

### 2.3. Methods

All patients with myocardial infarction studied were first examined clinically, undergoing a complete and complex evaluation according to the current standards, both in terms of anamnestic and objective examination, concentrating on examination of the cardiovascular system. Patients were monitored in the coronary intensive care unit for heart rate, ventricular rate, blood pressure, and oxygen saturation.

Laboratory tests were then undertaken as a matter of urgency, supplemented by subsequent data during hospitalization with the repeated measurement of certain biological parameters.

All patients with myocardial infarction underwent an electrocardiogram at the hospitalization stage, which was repeated during hospitalization and interpreted in dynamics.

After the initiation of emergency treatment and after clinical and paraclinical stabilization, the patients underwent echocardiography, and where there were indications of disease, they underwent exercise testing, 24-h Holter ECG monitoring, Doppler ultrasound of the peripheral arteries or carotids, myocardial scintigraphy, arteriography, coronary angiography, and measurement of myocardial revascularization. Lactic acid and cardiac markers were determined by spectrophotometric methods with an automatic biochemistry analyzer, Cobas model.

The measurement of carbonic anhydrase activity in red blood cells was performed using the stopped-flow method [20] with a HI-TECH SF-51MX rapid kinetics spectrophotometer provided with a RKBIN IS-1 kinetic program.

### 2.4. Statistical Analysis

The data were statistically analyzed using the EPIINFO application, version 6.0, a program of the Center for Disease Control and Prevention—CDC (Center of Disease Control and Prevention) in Atlanta, adapted to the processing of medical statistics. Frequency ranges, average parameter values and standard deviations were calculated. Tests of statistical significance by the χ2 method were used, and ANOVA (Brown–Forsythe) was used to compare the means. The level of statistical significance was 0.05.

## 3. Results

### 3.1. Characteristics of the Population (Age, Gender and Environment)

In our study group over 50% were women. The average age of the diabetic group (age range 37–73 years) was 56.7 ± 7.3 years, and of the non-diabetic group (age range 38–81 years) was 64.6 ± 6.9 years, and the patients came mainly from an urban environment (93.33%). This may be explained by the fact that diabetes is diagnosed faster and more easily in urban areas, while in rural areas there is still a lack of medical staff, a lack of education and a very low addressability of patients. Table 1 shows the characteristics of the diabetic and non-diabetic groups.

In the diabetic group, 93.3% come from urban areas and 6.66% from rural areas. In the non-diabetic group, 83.33% come from urban and 16.66% from rural areas. Their histories showed that over 55% of diabetic patients had had a diabetes diagnosis for less than 5 years, and the average duration of the disease in our study group was 5.9 years ± 1.3. Consistent with the mean low duration of their diabetes, 51.5% of patients had no complications.

The most common complications were macroangiopathic (stroke—5.4%, ischemic heart disease—18.8%, chronic obliterative arteriopathy—17.8%), followed by nephropathy (15.8%). Neuropathy and retinopathy were found in a small percentage of patients (6.4% and 4.0%, respectively), which were correlated with a shorter duration of diabetes in the patients studied. We observe a net favorable balance of macrovascular complications (stroke, obliterating arteriopathy, ischemic heart disease)—29.2%, compared to microangiopathic complications (nephropathy, neuropathy, retinopathy)—26.2%. There were patients in this study who had two or even three macrovascular complications without any microvascular complications.

### 3.2. Electrocardiographic Evaluation

In terms of the electrocardiographic evaluation at admission, the cases were distributed according to Table 2:

The electrocardiogram helped us to assess the topography and extent of the myocardial infarction. The most sensitive and specific ECG change for IMA is ST-segment elevation. Newly pathological Q waves show high sensitivity and specificity, with about 90% of patients having an evolving AMI. Pathological Q wave occurs later in the evolution of IMA than ST segment elevation. ST-segment elevation is sensitive in detecting myocardial ischemia, but cannot differentiate between AMI and unstable angina. T-wave changes can be induced by acute myocardial ischemia, in about one-third of patients developing AMI.

In our study, a significantly higher number of diabetic patients had a pathological Q wave at admission, which proves the delay in the presentation of these patients in an emergency department, and implicitly a delay in starting treatment. Non-diabetic patients who have more severe clinical manifestations than nondiabetics report to the emergency department earlier. ST-segment changes, which appear first in the evolution of an AMI, are found in a higher percentage in nondiabetic patients.

Therefore, the following locations were identified, Table 3:

In diabetic patients, the localization of AMI is mainly in the antero-septal and previously extended territory, these infarcts being large, and implicitly their evolution is unfavorable, being accompanied by multiple complications. This is due to the late presentation in the emergency services and implicitly due to the delay in starting the treatment.

Non-diabetic patients presented with smaller infarcts, located in the lower or lateral territory. The presence of large infarcts (previously extended, antero-septal) in diabetic patients and of smaller infarcts (lateral, anterior) in nondiabetic patients was observed.

### 3.3. Blood Sugar Values Distribution

The distribution of cases in terms of the blood sugar values at hospitalization is presented in Figure 1:

We considered the blood sugar to be pathological at a value of over 100 mg/dL according to the ADA standards [11]. In diabetics, the mean blood sugar at hospitalization was 248 mg/dL (with limits of 102 mg/dL and 572 mg/dL). 2.5% of diabetics had blood sugar values below 100 mg/dL, which indicates a good control of diabetes in these patients. 15.8% had blood sugar values between 100 and 140 mg/dL, 34.2% had glycemic values between 140 and 200 mg/dL, 39.4% had values between 200–400 mg/dL and 7.9% had values above 400 mg/dL.

In non-diabetic patients the blood sugar values were below the reference value of 100 mg/dL in 22 patients (75.5%), and in 8 patients the blood sugar value was over 100 mg/dL (with an upper limit of 140 mg/dL), but these values returned below 100 mg/dL in the next two days.

The enzymes of myocardial necrosis (Table 4), that served to confirm or exclude the diagnosis of major coronary heart disease determined in the two groups of patients were:creatine phosphokinase (CPK): VN = 25–90 U/I;creatine phosphokinase-MB (CPK-MB): VN < 5% of CK;lactic dehydrogenase (LDH): VN = 150–240 U/I;cardiac-specific troponins (cTnT and cTnI): VN < 0.1 ng/mL;oxaloacetic glutamic transaminase (GOT): VN < 35–40 IU.

CPK data for diabetic patients indicated that 76.2% of patients showed an increased value, with a mean of 458 U/I, and 23.8% had a normal value, with a mean of 68 U/I; this indicates that the latter patients had a major coronary accident more than 3–4 days ahead of other patients who had acute myocardial infarction (AMI) in the 3 days prior to the test.

CPK data for non-diabetic patients indicated that 88.7% of patients showed an increase above normal, with a mean value of 399 U/I, and the level was normal in 11.3% of patients with a mean value of 56 U/I.

CPK-MB data indicated that 78.7% of diabetic patients showed an increased value above normal, having a mean value of 68 U/I, while 21.3% of patients had a normal value with a mean of 5 U/I having a greater specificity for the AMI diagnosis.

CPK-MB data indicated that 91.2% of non-diabetic patients, having a mean value of 43 U/I and this was normal in 8.8% of patients with a mean value of 5 U/I.

LDH data presented an increased value in 97.5% of diabetic patients, with a mean value of 625 U/I, and in all non-diabetic patients with a mean value of 575 U/I.

The specific cardiac troponins determined showed elevated values above 0.1 ng/mL with an average value of 2.1 ng/mL.

GOT showed increased values in 73.8% of diabetic patients, with a mean value of 116 U/I, and in 59.8% of non-diabetic patients, with a mean value of 107 U/I.

In diabetics, severe myocardial ischemia may cause an increase in serum lactic acid (VN = 8–19.8 mg/dL) even without a decrease in blood pH. Based on these considerations, we tried to identify the place of lactic acid as a predictive marker of the evolution of diabetics with myocardial infarction. Lactic acid values were divided into 3 intervals: <25 mg/dL, 25–35 mg/dL and> 35 mg/dL (Table 5).

The main objective of analyzing biochemical markers in patients with acute chest pain according to modern standards is to stratify the risk of these patients. This means not only detecting or excluding myocardial necrosis, but also detecting patients at risk of developing a life-threatening cardiac event in the near future. In the present study, we tried to find new biochemical markers that would provide us with information on a future unfavorable evolution of IMA. Thus, we found that a higher number of diabetic patients had high lactic acid values compared to nondiabetics. We divided the high values of lactic acid into three intervals: <25 mg/dL; 25–35 mg/dL; and > 35 mg/dL.

In the diabetic patients group it was found that for 43.2% of patients the value of lactic acid increased significantly (>35 mg/dL), for 29.7% the value of lactic acid was between 25–35 mg/dL and for 27.0% the lactic acid value was below 25 mg/dL (*p* < 0.001).

In non-diabetics the value of lactic acid increased significantly for 36.7% of cases, for 33.3% the value of lactic acid was between 25–35 mg/dL and for 30% of cases the value of lactic acid was below 25 mg/dL (*p* < 0.001) (Table 5).

### 3.4. Correlation of Lactic Acid Values with ST-Segment Elevation

For diabetic patients with lactic acid values higher than 35 mg/dL, we found a regression below 30% of ST-segment elevation, without any obvious clinical improvement. In diabetic patients with lactic acid values between 25–35 mg/dL, we found a regression of 30–70% of ST-segment elevation, with obvious clinical improvement. There is also a linear correlation between the lactic acid value with ST elevation.

We made a correlation between the values of lactic acid and the regression of ST segment elevation. The regression (descent) of the ST segment elevation towards the isoelectric line represents a marker of favorable evolution of IMA, of repermeabilization of the hibernating areas around the myocardial necrosis. We took as a benchmark a regression of ST elevation with a percentage of 30%, on ECG performed in dynamics.

We found that diabetic patients who had high lactic acid values in the range >35 mg/dL had a small regression of ST segment elevation, by <30%, showing persistent ST segment elevation, leading to a wider necrotic area, and therefore to an unfavorable evolution of IMA.

Non-diabetic patients who had lower lactic acid values, <25 mg/dL, had a 100% regression of ST segment elevation by >30%, with a more favorable evolution and obvious clinical improvement (Table 6).

CA II activity showed higher values in diabetic patients compared to nondiabetic patients, while CA I activity was increased in both diabetic and nondiabetic patients, as shown in Table 7.

The biological parameters determined for patients from both groups are presented in Figure 2:

## 4. Discussion

The size of the infarction is crucial to survival. Diabetic patients have larger infarcts than non-diabetics, and since the size of myocardial infarction is closely correlated with cardiac performance and mortality, this may explain the higher mortality rate in diabetics. It has also been shown that women suffer from more severe heart attacks, which correlates well with the increase in mortality after acute myocardial infarction in diabetic women.

Diabetes is a major risk factor for the development of cardiovascular diseases. The risk of cardiovascular disease in patients with diabetes is two to three times higher than in patients without diabetes. In addition, diabetes is a strong risk factor for cardiovascular events after acute myocardial infarction [26]. Some studies indicate that diabetes is independently associated with impaired epicardial reperfusion and higher mortality [27].

Another factor associated with acute myocardial infarction that seems to be relevant in determining the increased mortality rate in diabetics is the location of the infarction. Diabetic patients most often have an anteroseptal localization of acute myocardial infarction, more commonly than non-diabetics, and mortality in such areas is higher than in any other localization [28,29].

It should be mentioned that if at hospitalization in most patients the ST segment changes and especially the ST segment elevation dominated, in both groups of diabetic patients and non-diabetic patients, at discharge the Q wave changes dominated, which overlaps with the evolution of the natural ECG of an acute myocardial infarction.

Echocardiography showed the presence of systolic dysfunction in a higher number of diabetic patients (71.3%) than non-diabetics (56.9%). The ejection fraction recorded lower values in diabetics than in non-diabetics (42.0%versus 47.5%) the difference being significant. This is consistent with some studies published in the literature [25]. The shortening fraction is higher in non-diabetic patients than in diabetics, but without significant differences between the two groups of patients (*p* > 0.05).

The analysis of paraclinical data performed on the groups of patients included in this study shows the presence of dyslipidemia (hypercholesterolemia, hypertriglyceridemia, hyperlipidemia, low HDL-cholesterol) in a higher number of diabetic patients compared to non-diabetic; and these dyslipidemias are more severe in diabetic than in non-diabetic patients. This data is consistent with the literature, given that the dyslipidemia profile of diabetic patients is frequently characterized by hypertriglyceridemia and decreased HDL-cholesterol; these dyslipidemias are independent cardiovascular risk factors in diabetic patients. In the studied groups, in the diabetic patients hypertriglyceridemia was present to a level of 55.5% and in non-diabetics to 20.1%. Low HDL-cholesterol was present in 44.6% of diabetics, compared to 23% in non-diabetics. This dyslipidemia profile of diabetic patients is more suitable for fibrate therapy (Helsinki-Heart study); however, many trials have shown that diabetics benefit equally with the non-diabetic population from statin therapy.

In the studied groups, we also observed a higher number of diabetic patients with hypercholesterolemia (47.5% versus 25.5%), so the dyslipidemia profile of the studied patients is different from the data published in the literature.

The study also focused on finding out whether there was an influence of lactic acid values on the evolution of patients with AMI. Thus, in diabetic patients we found an unfavorable evolution in those with high lactic acid values (>35 mg/dL). In patients with lactic acid above 35 mg/dL the risk of death was 1.7 times higher than in those with values between 25–35 mg/dL.

In non-diabetics, lethality rates in patients with lactic acid values above normal were lower than in diabetics (10.0% versus 18.2% at 25–35 mg/dL, and 22.2% versus 31.3% (>35 mg/dL), respectively.

It has also been observed that there is a direct correlation between CA II activity and diabetes, in the sense that diabetic patients have higher values of this isoenzyme, while CA I isoenzyme, known for its implications in vasoconstriction and vasodilation, has increased activity in all patients with myocardial infarction, regardless of their diabetic status [30,31,32]. The high values of CA II are also correlated with the high values of cardiac markers, meaning that the enzyme can also be an indicator of the state of damage to the heart tissue [33]. A long-term case follow-up study is needed to investigate the incidence of diabetes and cardiovascular complications of the disease [34].

## 5. Conclusions

Current studies indicate that there is no satisfactory biomarker that can specifically identify acute manifestations related to myocardial ischemia and its prognosis [31]. However, in this study the electrocardiogram analysis showed the presence in diabetics, especially, of much larger transmural myocardial infarctions and the fact that at the time of presentation to the doctor many diabetic patients had already developed a pathological Q wave of ECG necrosis. Lactic acid values together with the activity of CA II isoenzyme could be early markers in the prognosis and evolution of diabetic patients with acute myocardial infarction; their routine measurement may be included in the biological algorithm of acute myocardial infarction, but this should be fully researched in the future.

## Figures and Tables

**Figure 1 biology-11-01189-f001:**
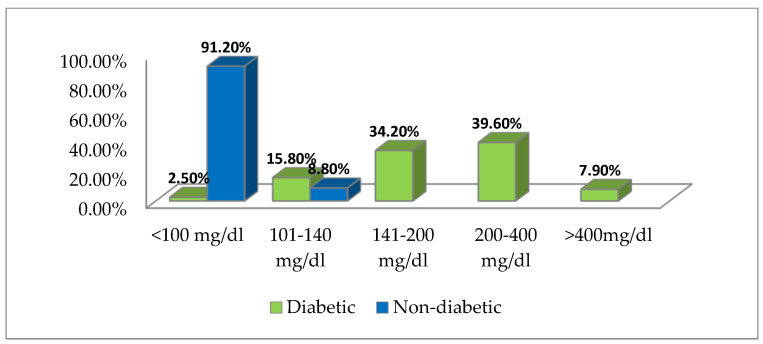
Case distribution according to blood sugar values at hospitalization.

**Figure 2 biology-11-01189-f002:**
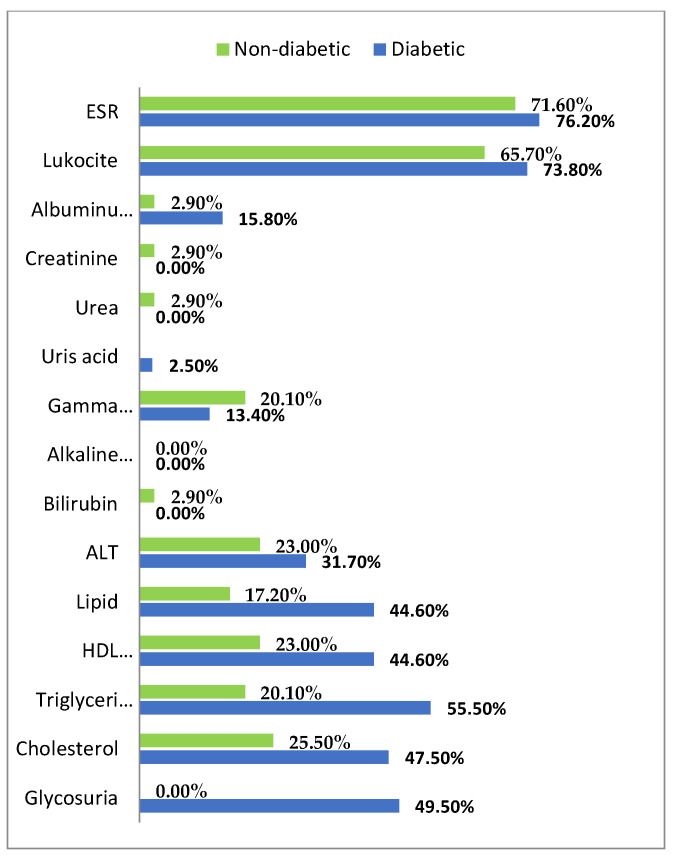
Distribution of cases according to biological parameters outside the limits of normality.

**Table 1 biology-11-01189-t001:** Characteristics of the population.

Baseline Characteristics of the Diabetic Group and Non-Diabetic Group	Diabetic MD ± DS		Non-Diabetic MD ± DS	
Age (years)	56.7 ± 7.3		64.6 ± 6.9	
Gender Women	19	Percentage% 63.33	17	Percentage% 56.66
Men	11	36.66	13	43.33
Environment Urban	28	93.33	25	83.33
Rural	2	6.66	5	16.66

**Table 2 biology-11-01189-t002:** Electrocardiographic evaluation case distribution at hospitalization.

Electrocardiographic Evaluation	Diabetic	Non-Diabetic
	%	%
Pathological Q wave	18.3	8.8
ST segment change	81.2	91.6
ST Elevation	78.7	82.8
ST Subleveling	2.5	8.8
T wave change	31.7	25.5

**Table 3 biology-11-01189-t003:** Distribution of cases according to the location of myocardial infarction.

Location of the Infarction	Diabetic	Non-Diabetic
	%	%
Lower	31.7	28.9
Previous	7.9	20.1
Anteroseptal	21.3	14.2
Previously stretched	18.3	5.9
Side	20.8	30.9

**Table 4 biology-11-01189-t004:** Enzymes of myocardial necrosis.

		Normal Values		Increased Values	*p* Value
Parameter	%	Mp ± DS	%	Mp ± DS	
Diabetic					
CPK	23.8	68.1 ± 8.5	76.2	458.5 ± 40.7	*p* < 0.001
CPK-MB	21.3	5.2 ± 1.3	78.7	68.2 ± 7.9	*p* < 0.01
LDH	2.5	159.7 ± 21.6	97.5	625.0 ± 64.3	*p* < 0.001
cTnT- cTnI	0	-	100	2.2 ± 0.3	-
GOT	26.2	27.5 ± 3.9	73.8	116.6 ± 12.3	*p* < 0.001
Non-diabetic					
CPK	11.3	56.3 ± 6.6	88.7	399.4 ± 41.3	*p* < 0.001
CPK-MB	8.8	5.0 ± 1.2	91.2	43.4 ± 5.7	*p* < 0.001
LDH	0	-	100	575.3 ± 61.8	-
cTnT- cTnI	0	-	100	2.1 ± 0.2	-
GOT	40.2	25.7 ± 3.2	59.8	107.2 ± 11.8	*p* < 0.01

Statistical significance: *p* < 0.001—highly significant; *p* < 0.01—very significant.

**Table 5 biology-11-01189-t005:** Lactic acid value.

Lactic Acid	Diabetic	Non-Diabetic	*p* Value
	%	%	
<25 mg/dL	27.0	30.0	*p* > 0.05
25–35 mg/dL	29.7	33.3	*p* < 0.01
>35 mg/dL	43.2	36.7	*p* < 0.001

**Table 6 biology-11-01189-t006:** Correlation of lactic acid values with ST-segment elevation.

Lactic Acid	Diabetic		Non-Diabetic	
ST-Segment Elevation	<30%	>30%	<30%	>30%
<25 mg/dL *p* > 0.05	20.0	80.0	-	100.0
25–35 mg/dL	36.4	63.6	20.0	80.0
>35 mg/dL *p* < 0.001	75.0	25.0	55.6	44.4

**Table 7 biology-11-01189-t007:** The activity of CA isozymes in diabetic and non-diabetic patients.

Red Blood Cells (UE/mL)	Normal Values	Diabetic	Non-Diabetic	*p* Value
CA I red blood cells (UE/mL)	0.262 ± 0.011	0.582 ± 0.021 *	0.574 ± 0.018 *	*p* < 0.01
CA II red blood cells (UE/mL)	1.015 ± 0.083	1.701 ± 0.118 *	1.042 ± 0.105	*p* < 0.05

* statistically significant difference compared to normal values (*p* < 0.05).

## Data Availability

Not applicable.

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
