# Peer review of "Correlation between Carbonic Anhydrase Isozymes and the Evolution of Myocardial Infarction in Diabetic Patients"

_biology, 2022, doi:10.3390/biology11081189_

Round 1
Reviewer 1 Report
Authors have sufficiently revised the manuscript. I have no further concerns.
Author Response
The authors acknowledge the useful observations and suggestions of the reviewer’s as concerns the manuscript entitled
Correlation between carbonic anhydrase isozymes and the evolution of myocardial infarction in diabetic patients by
Sorina Magheru1, Calin Magheru1, Florin Maghiar1, Liliana Sachelarie2*, Felicia Marc1, Corina Moldovan1, Laura Romila2, Anica Hoza1, Dorina Maria Farcas1, Irina Gradinaru3, Loredana Liliana Hurjui3
Thank you very much for your review,
Respectfully,
Prof.dr. Liliana Sachelarie

Reviewer 2 Report
The manuscript aimed to investigate the relationships between carbonic anhydrase isozymes and the evolution of AMI in diabetes. Diabetes and hyperglycemia beyond diabetes are common risk factors of adverse outcomes in AMI. On these grounds, I suggest you to describe this concept in the introduction to stress the importance of glycemic control; please find some reference to cite and discuss (in the intro): Association of stress induced hyperglycemia with angiographic findings and clinical outcomes in patients with ST-elevation myocardial infarction.
Diabetes Care. 2021 Nov;44(11):e192-e193. doi: 10.2337/dc21-0939. Epub 2021 Sep 16. PMID: 34531311
Predictive value of the stress hyperglycemia ratio in patients with acute ST-segment elevation myocardial infarction: insights from a multi-center observational study.
Am J Med Sci. 2022 Feb;363(2):122-129. doi: 10.1016/j.amjms.2021.06.025. Epub 2021 Sep 25. PMID: 34582805 The figure with the blood values is not updates since the cut off are 100 (not 110), 140 (not 150) and 200 mg/dl. Please modify accordingly and cite and discuss, in the methods, ADA standards of medical care: 6. Glycemic Targets: Standards of Medical Care in Diabetes-2022. Diabetes Care. 2022. PMID: 34964868
Author Response
The authors acknowledge the useful observations and suggestions of the academic editor’s as concerns the manuscript entitled
Correlation between carbonic anhydrase isozymes and the evolution of myocardial infarction in diabetic patients by
Sorina Magheru1, Calin Magheru1, Florin Maghiar1, Liliana Sachelarie2*, Felicia Marc1, Corina Moldovan1, Laura Romila2, Anica Hoza1, Dorina Maria Farcas1, Irina Gradinaru3, Loredana Liliana Hurjui3
According to the reviewer’s recommendations, the suggestions were carefully considered, as follows:
The physio-pathological processes that lead to the significant increase in cardiovascular damage in patients with diabetes are represented by three important biohumoral changes: hyperglycemia, hyperinsulinemia and the increase in circulating free fatty acids [8].
Hyperglycemia is the negative factor that influences the prognosis and clinical evolution of AMI, leading to higher mortality and more severe complications. Hyperglycemia is an important factor that determines endothelial dysfunction, leading to vascular damage and microvascular obstructions, and its effect of increasing oxidative stress, inflammation and platelet aggregation is also known. Thus, hyperglycemia causes a change in protein synthesis, with increased expression of pro-inflammatory factors (IL11, IL 6, TNF alpha, fibrinogen), oxidative stress and adhesion molecules (ICAM 1, VCAM 1), all of which result in endothelial damage and the appearance of atherosclerotic lesions [9,10].
Diabetes is accompanied by an extremely high incidence of acute cardiovascular events, 15-20% of patients presenting with acute myocardial infarction also have diabetes. The subsequent evolution of these patients is affected, the risk of adverse events being twice higher than in non-diabetics. The risk of cardiovascular disease is increased at values slightly above the allowed blood sugar limit and even at values above the normal limit. Hyperglycemia is associated with a poor prognosis in patients with acute myocardial infarction, regardless of the presence of diabetes [11,12].
The indicated articles were mentioned in the references.
Thank you very much for review reports and for the extremely useful observations and suggestions!
Respectfully,
Prof.dr. Liliana Sachelarie

Round 2
Reviewer 2 Report
The authors improved the manuscript as asked
This manuscript is a resubmission of an earlier submission. The following is a list of the peer review reports and author responses from that submission.
Round 1
Reviewer 1 Report
This study collected two groups of patients, namely diabetic patients with acute myocardial infarction and non-diabetic patients with acute myocardial infarction (n=30 patients each), and analyzed the lactic acid levels, carbonic anhydrase I/II activity in red blood cells, and correlated these parameters with electrocardiography. Authors concluded that lactic acid correlated with the activity of carbonic dioxide I isozyme could be early markers of the prognosis and evolution of diabetic patients with acute myocardial infarction. Overall, this is a very tedious study and has many major concerns.
1. In table 1, are these values percentage? If so, they should be written as 18.3 (not 18,3). The same problems also lie in other Tables. In addition, what does the percentage of 18.3% following Q wave mean? Does it mean 18.3% of the diabetic patients has pathologic Q wave? If so, it should be written as Pathological Q wave in the cell instead of a single Q wave.
2. For all the tables, the p values should be provided alongside the tables, either using the Student's t test or chi-square/fisher exact test. Simply listing out the percentages for the two groups does not have any scientific values.
3. Authors stated that a total of 30 patients were enrolled for each group. However, in Line 195, how would the non-diabetic group has 152 patients below 110mg/dl of blood sugar and 52 patients over 110mg/dl?
4. Table 3 and 5 are very very confusing and do not conform to the statements in the corresponding results sections. p values MUST be provided in the Tables.
5. From Table 6, CA II in red blood cells were observed to specifically increase in the diabetic group, instead of the CA I isozyme. How would author conclude that CA I together with lactic acid, is early marker for diabetic patients to develop myocardial infarction?
6. Discussion: authors should discuss the advantages of these markers, compared with other newly-identified markers such as endoreticulumn stress (ERS)-related secretory markers in early myocardial ischemia (for reference: Int J Legal Med. 2022 Jan;136(1):159-168; Front Cardiovasc Med. 2022 Mar 18;9:803532).
7. Conclusion should always be concise, preferentially limited in one paragraph.
8. Authors are encouraged to include more groups, including patients without development of myocardial infarction, with or without diabetes.
Minor concerns:
1. The format and writing style need extensive editing. For example, there should be spaces in many places (e.g. line 28 IIisozymeswere; line 159 diabeticgroup). Paragraph format should keep consistent (e.g. text-indent for all paragraph).
2. In Method section, authors should clearly state whether any ethical approval has been granted.
3. In method section, authors should state when the parameters were determined? immediately after admission to hospital? What type of samples were used for detection? (serum? whole blood, or plasma?). Why did authors detect the activity of CA isozymes in red blood cells?
4. The results section should be thoroughly revised. Some unnecessary information such as their blood glucose level, regions (rural or urban), duration of diabetes etc. should be concisely stated (may be incorporated into one Table so as to provide the basic demographic and clinical information for the studied groups of patients).
5. Authors stated that their aim is to correlate lactic acid level and CA isozymes with ST-segment. However, I did not see any correlation analysis. I suggest authors do linear correlation analysis. For example, since authors aimed to study myocardial infarction, authors can further provide patients' echocardiograph data and do linear correlation between lactic acid, CA isozymes with the myocardial infarction indices (cardiac enzymes CK-MB for example, electrocardiograph data, and echocardiograph data).
6. The conclusion "Lactic acid correlated with the activity of I isozyme of carbonic dioxide which could be early markers of the prognosis and evolution of diabetic patients with acute myocardial infarction" is very very confusing. Do authors mean "lactic acid together with the activity of CA isozyme"?
7. Serious language polish and formatting is badly needed.
Author Response
The authors acknowledge the useful observations and suggestions of the reviewer’s as concerns the manuscript entitled first
Correlation between carbonic anhydrase isozymes and the evolution of myocardial infarction in diabetic patients by
Sorina Magheru1, Calin Magheru1, Florin Maghiar1, Liliana Sachelarie2*, Felicia Marc1, Corina Moldovan1, Laura Romila2, Anica Hoza1, Dorina Maria Farcas1, Irina Gradinaru3, Loredana Liliana Hurjui3
According to the reviewer’s recommendations, the suggestions were carefully considered, as follows:
This study collected two groups of patients, namely diabetic patients with acute myocardial infarction and non-diabetic patients with acute myocardial infarction (n=30 patients each), and analyzed the lactic acid levels, carbonic anhydrase I/II activity in red blood cells, and correlated these parameters with electrocardiography. Authors concluded that lactic acid correlated with the activity of carbonic dioxide I isozyme could be early markers of the prognosis and evolution of diabetic patients with acute myocardial infarction. Overall, this is a very tedious study and has many major concerns.
- In table 1, are these values percentage? If so, they should be written as 18.3 (not 18,3).
Done.
The same problems also lie in other Tables.
Done.
In addition, what does the percentage of 18.3% following Q wave mean? Does it mean 18.3% of the diabetic patients has pathologic Q wave? YES.
If so, it should be written as Pathological Q wave in the cell instead of a single Q wave. Done
- For all the tables, the p values should be provided alongside the tables, either using the Student's t test or chi-square/fisher exact test. Simply listing out the percentages for the two groups does not have any scientific values.
The changes have been made.
- Authors stated that a total of 30 patients were enrolled for each group. However, in Line 195, how would the non-diabetic group has 22 patients below 110mg/dl of blood sugar and 8 patients over 110mg/dl. It was a transcription error.
- Table 3 and 5 are very very confusing and do not conform to the statements in the corresponding results sections. p values MUST be provided in the Tables.
The changes been made.
- From Table 6, CA II in red blood cells were observed to specifically increase in the diabetic group, instead of the CA I isozyme. How would author conclude that CA I together with lactic acid, is early marker for diabetic patients to develop myocardial infarction?
It was corrected in conclusions, correlation being between CA II and lactic acid, with myocardial infarction.
- Discussion: authors should discuss the advantages of these markers, compared with other newly-identified markers such as endoreticulumn stress (ERS)-related secretory markers in early myocardial ischemia (for reference: Int J Legal Med. 2022 Jan;136(1):159-168; Front Cardiovasc Med. 2022 Mar 18;9:803532).
In this study we tried to identify biomarkers who can indicate us the degree of degradation of the myocardial tissue and the poor evolution of myocardial infarction. Indeed there are advantages of these markers and the goal of their study was similar.
- Conclusion should always be concise, preferentially limited in one paragraph. We tried to keep in conclusions all the datas that we studied
- Authors are encouraged to include more groups, including patients without development of myocardial infarction, with or without diabetes. Thank you for your advice perhaps in a future study we will investigate this.
Minor concerns:
- The format and writing style need extensive editing. For example, there should be spaces in many places (e.g. line 28 IIisozymes were; line 159 diabetic group). Paragraph format should keep consistent (e.g. text-indent for all paragraph). Done
- In Method section, authors should clearly state whether any ethical approval has been granted.
It is written at the end just before references
- In method section, authors should state when the parameters were determined? immediately after admission to hospital? What type of samples were used for detection? (serum? whole blood, or plasma?). Why did authors detect the activity of CA isozymes in red blood cells?
Yes, immediately after admission, from whole blood, because the CA is found in red blood cells.
- The results section should be thoroughly revised. Some unnecessary information such as their blood glucose level, regions (rural or urban), duration of diabetes etc. should be concisely stated (may be incorporated into one Table so as to provide the basic demographic and clinical information for the studied groups of patients).
OK, but some of the information are necessary for the cardiologist, for the evolution of the myocardial infarction such as blood glucose level and the duration of the diabetes.
- Authors stated that their aim is to correlate lactic acid level and CA isozymes with ST-segment. However, I did not see any correlation analysis. I suggest authors do linear correlation analysis. For example, since authors aimed to study myocardial infarction, authors can further provide patients' echocardiograph data and do linear correlation between lactic acid, CA isozymes with the myocardial infarction indices (cardiac enzymes CK-MB for example, electrocardiograph data, and echocardiograph data).
The size of the ST segment (which is electrocardiograph data) was correlated with the level of lactic acid and CA isozymes, trying to correlate the value of these biomarkers with the severity of the myocardial infarction.
- The conclusion "Lactic acid correlated with the activity of I isozyme of carbonic dioxide which could be early markers of the prognosis and evolution of diabetic patients with acute myocardial infarction" is very very confusing. Do authors mean "lactic acid together with the activity of CA isozyme"?
Yes together but it is also possible separately.
- Serious language polish and formatting is badly needed.
We checked the article from the point of view of languages errors and grammatical errors.
Thank you very much for your extremely useful observations and suggestions!
Prof.dr. Liliana Sachelarie

Reviewer 2 Report
Magheru and colleagues performed an interesting analysis. The study showed that CA isozyme seem has a role in myocardial infarction, even more in diabetic patients. However, the study is not substantial enough to be published in this journal. The study has a strong statistical component but leaves out relevant parameters (e.g. gender). On the other hands, further analysis are needed to make the claim/conclusion on wich the manuscript focuses. Although the paper is well thought out and the results are significant, I do not believe that the study is sufficiently novel or relevant.
Author Response
The authors acknowledge the useful observations and suggestions of the reviewer’s as concerns the manuscript entitled first
Correlation between carbonic anhydrase isozymes and the evolution of myocardial infarction in diabetic patients by
Sorina Magheru1, Calin Magheru1, Florin Maghiar1, Liliana Sachelarie2*, Felicia Marc1, Corina Moldovan1, Laura Romila2, Anica Hoza1, Dorina Maria Farcas1, Irina Gradinaru3, Loredana Liliana Hurjui3
According to the reviewer’s recommendations, the suggestions were carefully considered, as follows:
Thank you for appreciating the article.
In our study, we wanted to highlight and to establish whether there was a correlation between the evolution of myocardial infarction in diabetic patients and the activity of CA isoenzymes.
The study did not focus on gender but on diabetic patients with acute myocardial infarction, and non-diabetic patients with acute myocardial infarction.
The study required rigorous and long-term research and the results demonstrate what we set out to do.
We consider that the study is of interest in the context of the increase in the number of acute myocardial infarction cases related to diabetes. Current research focuses on the discovery of new biomarkers that highly and early highlight myocardial ischemia.
We made the required changes.
Thank you again for your appreciation!
Prof.dr. Liliana Sachelarie

Round 2
Reviewer 1 Report
Authors have revised their manuscript to some extent. I still have the following concerns.
1. Tables should be three-line, and all the p values mentioned in the RESULTS sections should be written out in the TABLES, preferentially at the right columns.
2. The descriptions are not consistent with what authors presented in Table 5.
3. Figure 1 needs revision. For example, Y-axis should have caption.
4. The format and writing style still need extensive editing. For example, it should be text-indent for the first line of paragraphs (e.g. Line 200, 204, 239, 285).
5. Authors should include a statement of limitations since they did not include more groups. Also, authors should mention the limitations of their markers, as compared with other newly identified biomarkers, since they did not do stability tests of their markers.
Author Response
The authors acknowledge the useful observations and suggestions of the reviewer’s as concerns the manuscript entitled
Correlation between carbonic anhydrase isozymes and the evolution of myocardial infarction in diabetic patients by
Sorina Magheru1, Calin Magheru1, Florin Maghiar1, Liliana Sachelarie2*, Felicia Marc1, Corina Moldovan1, Laura Romila2, Anica Hoza1, Dorina Maria Farcas1, Irina Gradinaru3, Loredana Liliana Hurjui3
According to the reviewer’s recommendations, the suggestions were carefully considered, as follows:
Authors have revised their manuscript to some extent. I still have the following concerns.
- Tables should be three-line, and all the p values mentioned in the RESULTS sections should be written out in the TABLES, preferentially at the right columns.
Done
Table 4. Enzymes of myocardial necrosis
|
|
|
Normal values |
|
Increased values |
P value |
|
Parameter |
% |
Mp±DS |
% |
Mp±DS |
|
|
Diabetic |
|
|
|
|
|
|
CPK |
23.8 |
68,1±8,5 |
76.2 |
458,5±40,7 |
p<0.001 |
|
CPK-MB |
21.3 |
5,2±1,3 |
78.7 |
68,2±7,9 |
p<0.01 |
|
LDH |
2.5 |
159,7±21,6 |
97.5 |
625,0±64,3 |
p<0.001 |
|
cTnT- cTnI |
0 |
- |
100 |
2,2±0,3 |
- |
|
GOT |
26.2 |
27,5±3,9 |
73.8 |
116,6±12,3 |
p<0.001 |
|
Non-diabetic |
|
|
|
|
|
|
CPK |
11.3 |
56,3±6,6 |
88.7 |
399,4±41,3 |
p<0.001 |
|
CPK-MB |
8.8 |
5,0±1,2 |
91.2 |
43,4±5,7 |
p<0.001 |
|
LDH |
0 |
- |
100 |
575,3±61,8 |
- |
|
cTnT- cTnI |
0 |
- |
100 |
2,1±0,2 |
- |
|
GOT |
40.2 |
25,7±3,2 |
59.8 |
107,2±11,8 |
p<0.01 |
Table 5. Lactic acid value
|
Lactic acid |
Diabetic |
Non-diabetic |
P value |
|
% |
% |
|
|
|
<25 mg/dl |
27.0 |
30.0 |
P> 0.05 |
|
25-35 mg/dl |
29.7 |
33.3 |
p<0.01 |
|
>35 mg/dl |
43.2 |
36.7 |
p<0.001 |
Tabel 7. The activity of CA isozymes in diabetic and non-diabetic patients
|
Red blood cells (UE/ml) |
Normal values |
Diabetic |
Non-diabetic |
P value |
|
CA I red blood cells (UE/ml) |
0.262±0.011 |
0.582±0.021* |
0.574±0.018* |
p<0.01 |
|
CA II red blood cells (UE/ml) |
1.015±0.083 |
1.701±0.118* |
1.042±0.105 |
p<0.05 |
* statistically significant difference compared to normal values (p <0.05)
- The descriptions are not consistent with what authors presented in Table 5.
The main objective of analyzing biochemical markers in patients with acute chest pain according to modern standards is to stratify the risk of these patients. This means not only detecting or excluding myocardial necrosis, but also detecting patients at risk of developing a life-threatening cardiac event in the near future. In the present study, we tried to find new biochemical markers that would provide us with information on a future unfavorable evolution of IMA. Thus, we found that a higher number of diabetic patients had high lactic acid values ​​compared to nondiabetics. We divided the high values ​​of lactic acid into three intervals: <25 mg / dl; 25-35 mg / dl; and > 35 mg / dl. We made a correlation between the values ​​of lactic acid and the regression of ST segment elevation. The regression (descent) of the ST segment elevation towards the isoelectric line represents a marker of favorable evolution of IMA, of repermeabilization of the hibernating areas around the myocardial necrosis. We took as a benchmark a regression of ST elevation with a percentage of 30%, on ECG performed in dynamics.
In diabetic patients, who had high lactic acid values ​​in the range> 35 mg / dl, it was a small regression of ST segment elevation, by <30%, thus persisting for a longer time ST segment elevation, leading to a wider necrotic area, and therefore to an unfavorable evolution of IMA.
Non-diabetic patients who had lower lactic acid values, <25 mg / dl, had a 100% regression of ST segment elevation by> 30%, with a more favorable evolution and obvious clinical improvement.
In the diabetic patients group it was found that for 43.2% of patients the value of lactic acid increased significantly (> 35 mg/dl), for 29.7% the value of lactic acid was between 25-35 mg/dl and for 27.0% the lactic acid value was below 25 mg/dl (p<0.001).
In non-diabetics the value of lactic acid increased significantly for 36.7% of cases, for 33.3% the value of lactic acid was between 25-35 mg / dl and for 30% of cases the value of lactic acid was below 25 mg/dl, (p<0.001) Table 5.
- Figure 1 needs revision. For example, Y-axis should have caption.
We deleted the figure 1and insert table nr.1
- Characteristics of the population (age, gender and environment)
Tabel 1 Characteristics of the population
|
Baseline Characteristics of the Diabetic Group and Non-diabetic Group |
Diabetic MD±DS |
Non-diabetic MD±DS |
||
|
Age (years) |
56.7±7.3 |
64.6±6.9 |
||
|
Gender Women |
19 |
Percentage% 63.33 |
17 |
Percentage% 56.66 |
|
Men |
11 |
36.66 |
13 |
43.33 |
|
Environment Urban |
28 |
93.33 |
25 |
83.33 |
|
Rural |
2 |
6.66 |
5 |
16.66 |
In our study group over 50% were women, the average age of the diabetic group (age between 37-73 years old) was 56.7±7.3 years, and of the non-diabetic group (age between 38- 81 years old) was 64.6±6.9 years, and the patients came mainly from the urban environment (93.33%). This is explained by the fact that diabetes is diagnosed faster and easier in urban areas, while in rural areas there is still a lack of medical staff, a lack of education and a very low addressability of patients, table1.
- The format and writing style still need extensive editing. For example, it should be text-indent for the first line of paragraphs (e.g. Line 200, 204, 239, 285).
Done
- Authors should include a statement of limitations since they did not include more groups. Also, authors should mention the limitations of their markers, as compared with other newly identified biomarkers, since they did not do stability tests of their markers.
Current studies indicate that there is no satisfactory biomarker that can specifically identify acute manifestations related to myocardial ischemia and its prognosis. [27]. However, in this study the electrocardiogram analysis showed the presence in diabetics, especially of much larger transmural myocardial infarctions and the fact that at the time of presentation to the doctor many diabetic patients had already developed pathological Q wave of ECG necrosis. Lactic acid correlated with the activity of CA II isoenzyme could be early markers in the prognosis and evolution of diabetic patients with acute myocardial infarction, their routine measurement may be included in the biological algorithm of acute myocardial infarction, but this should be fully researched in the future
Thank you very much for review reports and for the extremely useful observations and suggestions!
Kind regards,
Prof.dr. Liliana Sachelarie

Reviewer 2 Report
The manuscript has not been significantly modified. This study should provide more relevant and novel results, a more scientifically based introduction and discussion, and a more extensive experimental design for publication in this journal.
Author Response
We have made changes to the manuscript and please consider our work. We have attached the manuscript in which we have introduced everything you mentioned in the first revision and everything that the other reviewers have suggested.
Thank you very much for your review
Respectfully,
Prof.dr. Liliana Sachelarie
